# Strengthening Kangaroo Mother Care at a tertiary level hospital in Zambia: A prospective descriptive study

Nobutu Muttau[1], Martha Mwendafilumba[1], Branishka Lewis[1‡], Keilya Kasprzyk[1,2‡], Colm Travers[3], J. Anitha Menon[1,4], Kunda Mutesu-Kapembwa[5], Aaron Mangangu[1☉], Herbert Kapesa[1☉], Albert Manasyan[1,3]*

1 Department of Reproductive, Maternal, Newborn, and Child Health, Centre for Infectious Disease Research in Zambia, Lusaka, Zambia, 2 The Brown School, Washington University in St. Louis, St. Louis, Missouri, United States of America, 3 Department of Pediatrics, Division of Neonatology, University of Alabama at Birmingham, Birmingham, Alabama, United States of America, 4 Department of Psychology, University of Zambia, Lusaka, Zambia, 5 Department of Neonatology, Women and Newborn Hospital, University Teaching Hospital, Lusaka, Zambia

☉ These authors contributed equally to this work.
‡ BL and KK also contributed equally to this work.
* albertmanasyan@uabmc.edu

**Data Availability Statement:** All data is freely available upon request to the Senior Author of this paper. Data generated from this study is owned by the Ministry of Health of the Republic of Zambia thus will be shared upon request from the

## Abstract

### Background

Globally, complications due to preterm birth are the leading contributor to neonatal mortality, resulting in an estimated one million deaths annually. Kangaroo Mother Care (KMC) has been endorsed by the World Health Organisation as a low cost, safe, and effective intervention in reducing morbidity and mortality among preterm infants. The objective of this study was to describe the implementation of a KMC model among preterm infants and its impact on neonatal outcomes at a tertiary level hospital in Lusaka, Zambia.

### Methods

We conducted a prospective descriptive study using data collected from the KMC room at the University Teaching Hospital between January 2016 and September 2017. Mothers and government nurses were trained in KMC. We monitored skin-to-skin and breastfeeding practices, weight at admission, discharge, and length of admission.

### Results

We enrolled 573 neonates into the study. Thirteen extremely low weight infants admitted to the KMC room had graduated to Group A (1,000g-1,499g) at discharge, with a median weight gain of 500g. Of the 419 very low weight neonates at admission, 290 remained in Group A while 129 improved to Group B (1,500g-2,499g), with a median weight gain of 280g. Among the 89 low weight neonates, 1 regressed to Group A, 77 remained in Group B, and 11 improved to Group C (≥2,500g), individually gaining a median of 100g. Of the seven

corresponding author or Dr Sylvia Machona-Muyunda (Head of the NICU, University Teaching Hospital, Ministry of Health: sylviammm@yahoo.com).

**Funding:** Research reported in this publication was supported by a grant from The ELMA Foundation (14-F0023). The funders had no role in study design, data collection and analysis, decision to publish, or preparation of the manuscript.

**Competing interests:** The authors have declared that no competing interests exist.

normal weight neonates, 6 remained in Group C individually gaining a median of 100g, and 1 regressed to Group B. Among all infants enrolled, two (0.35%) died in the KMC room.

## Conclusions

Based on the RE-AIM metrics, our results show that KMC is a feasible intervention that can improve neonatal outcomes among preterm infants in Zambia. The study findings show a promising, practical approach to scaling up KMC in Zambia.

## Trial registration

The trial is registered under ClinicalTrials.gov under the following ID number: NCT03923023.

## Introduction

Globally, 2.5 million newborns die every year, contributing to 47% of under-5 child deaths [1]. Complications due to preterm birth account for 35% of neonatal deaths [2]. Sub-Saharan Africa is among the regions with the highest preterm birth rates, accounting for 60% of the annual 15 million preterm births worldwide [3]. Premature infants are at a greater risk of dying early from serious health conditions such as hypothermia, sepsis, respiratory and feeding problems [4]. They require advanced neonatal care such as incubators and ventilator machines, which are often unavailable in low-resource settings due to their high costs [4]. Over 50% of neonatal deaths among preterm and/or low birthweight infants could be averted through simple evidence-based interventions at the time of birth [5]. These interventions can be adapted and integrated into the existing health systems to improve newborn survival in settings where conventional care is limited [6, 7].

The World Health Organisation (WHO) in 2015, recommended that kangaroo mother care (KMC) be practiced continuously on all clinically stable preterm infants in all settings [8]. KMC is a proven, low-cost, and effective intervention that was developed by Dr. E. Rey in 1978 for preterm (born before 37 weeks of gestation) and/or low birthweight infants (<2500g) [9–11]. It is a method for parents or guardians to provide warmth to their newborns through continuous skin-to-skin contact, in which the newborn is placed (naked) in a reclined position between the mother or guardians' breasts and covered with a blanket [13, 37]. The KMC position encourages early and exclusive breastfeeding, infant weight gain as well as promotes mother-infant bonding [12].

A meta-analysis demonstrated that KMC contributes to a 51% reduction of neonatal deaths among preterm and/or low birthweight infants [13]. In comparison to standard conventional care, it has also been found to reduce the duration of hospital stay and neonatal morbidity primarily from infection [11]. Numerous studies have explored the facilitators and challenges of implementing KMC in hospitals in different settings [14–18]. Strong local leadership, availability of well-equipped KMC units, trained personnel, and supplies were shown to support effective KMC implementation in Africa [14, 16, 17]. Notably, family unreadiness and the absence of fully functional referral systems were identified as significant barriers in Indonesia and Pakistan, respectively [15, 19].

Similar to other sub-Saharan countries, high-quality neonatal care in Zambia remains limited due to shortages in staff, clinical supplies, and equipment [20, 21]. Almost 23 in every 1000

babies born die annually before the age of five [22, 23]. Although KMC was officially introduced in Zambia in 2010, its continued practice was not sustained in the following years [24]. In response to the 2015 WHO guidelines, the Zambian government has now prioritized high-impact interventions such as KMC as program implementation areas in need of scale-up [8]. Its commitment to addressing the top major causes of neonatal mortality in the country led to the development of several national guidelines and policies, including the Kangaroo Mother Care National Guidelines 2018 [25]. The objective of this study was to describe the implementation of a KMC model among preterm infants and its impact on neonatal outcomes at a tertiary level hospital in Lusaka, Zambia. Furthermore, this manuscript describes the operational aspects of the implementation of the KMC model through the five evaluation dimensions of the RE-AIM framework (reach, effectiveness, adoption, implementation, and maintenance) [26, 27].

## Methods

This prospective descriptive study was nested within a larger study entitled "Preterm Resources, Education and Effective Management for Infants" (PREEMI). PREEMI was an initiative of four key collaborators: the Centre for Infectious Disease Research in Zambia (CIDRZ), the University of Alabama at Birmingham, the Zambian Ministry of Health (MoH), and the Zambian Ministry of Community Development, Mother and Child Health. The parent study aimed to reduce neonatal mortality among preterm infants by introducing the PREEMI package using a stepped wedge design in three public health facilities within Lusaka, Zambia, between May 2015 to September 2017. PREEMI was a quality improvement study using a hybrid training and onsite coaching and mentorship approach, rolled out through a stepped wedge design. The study used quantitative and qualitative methods to assess its impact at first level hospitals; however, in this manuscript, we describe the work done in a KMC room at a tertiary level hospital.

As part of the study, a KMC room was refurbished at the University Teaching Hospital (UTH), the largest and main public tertiary level hospital in Lusaka, Zambia. UTH serves as the national referral hospital and receives a high volume of referrals from both public and private healthcare institutions across the country [28]. The hospital also provides primary level care to approximately 2 million people within the capital and in this capacity, assisted with the strengthening of KMC services at the hospital [28]. At the time of our study, UTH conducted on average 20,000 deliveries per year, with a preterm birth rate of 23% and institutional neonatal mortality rate of 46.5% [29]. The Neonatal Intensive Care Unit (NICU) had 20 working incubators and a bed capacity of 60; however, its occupancy rate was between 100% to 185%. Due to overcrowding, nosocomial infections, and limited space in the NICU, the KMC room was established as an intermediary room between the NICU and home. It enabled newborns not requiring NICU services but who needed further monitoring to stay at the hospital before hospital discharge, consequently decongesting the NICU. Prior to the establishment of the KMC room, all newborns who did not require further NICU care but required further monitoring prior to hospital discharge were all kept in the NICU. All newborns discharged from the NICU and admitted to the KMC room between January 2016, and September 2017 were included in the study. The KMC room inclusion criteria were newborns (a) deemed stable by the neonatologist (i.e., breathing spontaneously without additional oxygen, hemodynamically stable, feeding orally), (b) who did not require further NICU care but required further clinical monitoring within the hospital, and (c) mothers who were willing to stay in the KMC room and be taught KMC techniques.

### Intervention

The KMC model was embedded within the routine standard of care and implemented by a study team working closely with the Ministry of Health (MoH) staff at UTH. Our study

refurbished the KMC room and equipped the 10-bedded room with the necessary equipment and supplies. These included 1 television set with 28 KMC educational videos, 2 infant weighing scales, penguin newborn suction (Laerdal Global Health, Stavanger, Norway), bag and mask resuscitators, Nifty feeding cups (Laerdal Global Health, Stavanger, Norway), bed linen, hand sanitizers, locker, gloves, chairs, syringes, mothers' gowns and bed slippers, room heaters, and KMC registers [30]. The educational videos were shown throughout the day, except in the night, in the local dialects (Bemba, Nyanja and Lozi) and English. The videos covered various topics that included KMC, breastfeeding, and newborn danger signs. All educational sessions were facilitated by both the study QI and the government nurse in the KMC room.

The study team consisted of a quality improvement nurse (QI), data coordinator and data associate working under the implementing partner, the Centre for Infectious Disease Research in Zambia, while the Zambian Ministry of Health staff included government nurses and community Safe Motherhood Action Groups (SMAGs). Study implementation was coordinated by the study QI nurse, who worked alongside several government nurses, including student nurses owing to limited staff, in delivering nursing care for 8 to 10 hour shifts, twenty-four hours a day in the KMC room. Throughout the study, the QI nurse served as a change champion in providing daily onsite coaching, mentorship, and monitoring to both the government nurses and mothers on KMC ensuring (1) continuous skin-to-skin contact (2) exclusive breastfeeding, and (3) follow up at home after discharge [31]. Government nurses managed the KMC room and ensured the SMAGs carried out community-level follow up of all newborns after discharge from the hospital. In addition to providing capacity building and supplies, the study also displayed information, education, and communication (IEC) materials in different sections of the KMC room.

The study QI nurse collected daily individual-level baseline demographic and clinical data of both the mother and the infant from the KMC registers. Data were collected prospectively as part of the standard of care routine procedure and entered into the study-specific database. The KMC register included demographic characteristics such as infant's date of birth, sex, weight at admission to the KMC room and discharge, and flowback. The clinical variables of importance were the infants' gestational age (GA), cause of admission into the NICU (i.e. prematurity, low birthweight, sepsis, respiratory distress, etc), admission and discharge weight (grams) and diet (exclusive breastmilk/mixed). Other variables of importance included discharge data (date and reason for discharge from KMC), date of admission, and duration of stay in the KMC room.

## Conceptual framework

We sought to assess the KMC model using the RE-AIM framework developed to guide evaluations of different public health interventions and programs through its five dimensions: Reach, Effectiveness, Adoption, Implementation and Maintenance (Fig 1) [32]. RE-AIM goes beyond standard measures of efficacy and effectiveness by answering practical questions on whether implementation strategies are helpful and can be readily accepted, widely implemented and sustained at individual and organisational level [26, 33]. Following this framework, we defined 'Reach' as the proportion of newborns discharged from the NICU and eligible to be admitted to the KMC room. 'Effectiveness' was considered the KMC model's impact on neonatal outcomes, including the quality of life, adverse effects and economic outcomes. 'Adoption' was the proportion of mothers, government nurses and SMAGs willing to initiate and practice KMC. 'Implementation' refers to how the KMC model components were delivered, including the time and cost implications. Lastly, 'Maintenance' was how our KMC model was institutionalised or made part of the routine standard of care practice and policy.

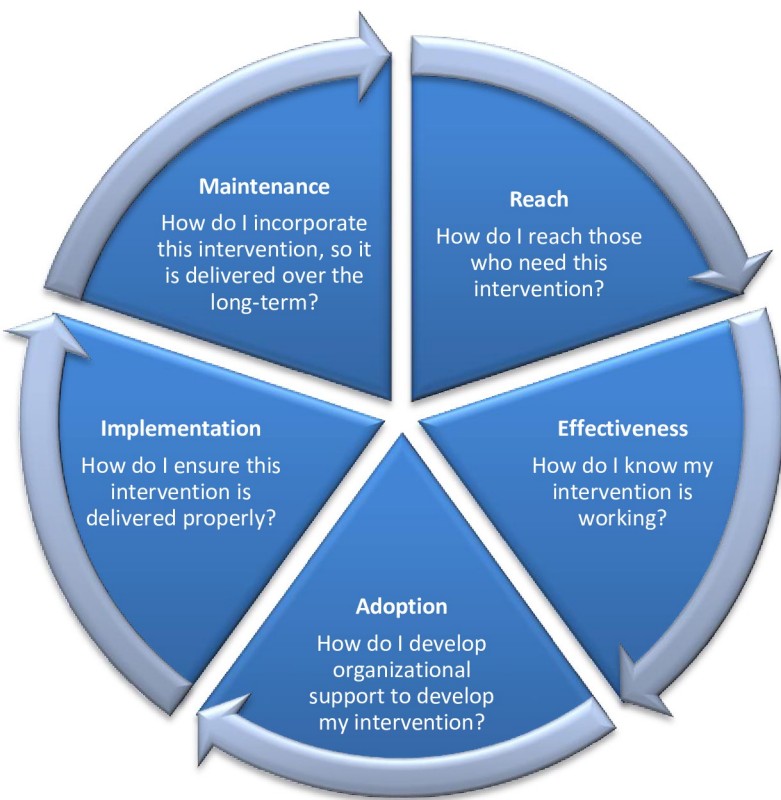

**Fig 1. Schematic figure depicting the RE-AIM framework.**

## Statistical analysis

Data were entered centrally into the OpenClinica database and checked for errors and inconsistencies on a routine basis by the trained data associate. Quality assurance and quality control (QA/QC) was initially done at UTH by the QI nurse and then at CIDRZ by the data associate before data entry. The Data Coordinator generated monthly data reports and sent them to the hospital for error resolutions. Study participants were given unique patient identifiers; thus, no personal details that could identify participants were entered into the database. Continuous variables such as birth weight and duration of stay were summarised and presented as medians and ranges. The categorical variables were presented as proportions. Comparisons between birth weight groups were performed using Chi-square and Kruskal Wallis tests; for this analysis $p < 0.05$ was considered statistically significant. Additionally, descriptive statistics were collected and displayed using frequencies. Non-normal data were represented using the median and interquartile range. Participants with missing data for either weight at admission or weight at discharge from the KMC room were omitted from weight analysis. Data were analyzed using STATA version 15–1 (Stata Corp, College Station, TX, USA).

## Ethical considerations

The study was approved by the University of Zambia Biomedical Research Ethics Committee (UNZA BREC–Ref. 012-12-16), the Zambia National Health Research Authority (ZNHRA), and the University of Alabama of Birmingham (UAB-IRB: 300000669). The Zambian Ministry of Health granted permission to publish these study findings. The institutional review boards

waived the requirement to obtain informed consent from all study participants as KMC was the standard of care.

## Results

### Reach

The study assessed participant representation based on the total number of infants in the NICU at UTH and potentially eligible for discharge to the KMC room between January 2016 and September 2017. We calculated the proportion of newborns that took part in the study based on the total number of infants that were discharged from the NICU and eligible to be admitted to the KMC room. During this period, all 573 mother-infant pairs enrolled in this study were eligible and willing to be admitted to the UTH KMC room (2016–293; 2017–280).

Table 1 shows the demographic characteristics of the neonates included in the study. The majority of neonates were female (60.91%), and all of the study participants were exclusively breastfed. Data on HIV exposure was available for almost half of the study participants (54.97%), with 69 (12.04%) having known HIV exposure and 246 (42.93%) being HIV unexposed. Upon discharge from the KMC room, 536 (93.54%) were discharged home, with 30 neonates (5.24%) readmitted to the NICU, defined as flowback (Table 1). Of the 30 neonates, 28 were readmitted to the NICU directly from the KMC room, with only two readmitted to the NICU after being discharged home. Five infants (0.87%) left the KMC room against medical advice (LAMA). Throughout the study, two infants (0.35%) died while in the KMC room. One neonate suffered from recurrent apnoeic spells and died 2 days after being admitted into the KMC room, while the other suffocated in her sleep 10 days after being admitted into the KMC room.

A total of 26 neonatal conditions were listed as the primary cause of NICU admission for neonates admitted to the KMC room, with four neonates (0.70%) missing cause for admission. Table 2 shows the frequencies of the nine most common conditions with the rest categorized under "other" and "missing". The most common neonatal conditions were prematurity (33.74%), very low birthweight (VLBW) (28.47%), low birthweight (LBW) (18.63%),

**Table 1. Demographic characteristics of neonates (n = 573).**

| Demographic Characteristics | % (n) |
|---|---|
| **Sex** | |
| Female | 60.91% (349) |
| **Diet** | |
| **Exclusive Breast Milk** | 100% (573) |
| **HIV Exposure Status** | |
| **Exposed** | 12.04% (69) |
| **Not Exposed** | 42.93% (246) |
| **Missing** | 45.03% (258) |
| **Birthweight** | |
| **<2,500g** | 97.1% (565) |
| **≥2,500g** | 2.9% (8) |
| **Outcome** | |
| **Home** | 93.54% (536) |
| **Flowback** | 5.24% (30) |
| **LAMA** | 0.87% (5) |
| **Died** | 0.35% (2) |

**Table 2. Primary cause of NICU admission for neonates (n = 573).**

| Primary Cause of NICU Admission | % (n) |
|---|---|
| **Prematurity** | 33.74 (192) |
| **VLBW** | 28.47 (162) |
| **LBW** | 18.63 (106) |
| **RDS** | 4.57 (26) |
| **Sepsis** | 3.34 (19) |
| **ELBW** | 2.11 (12) |
| **Jaundice** | 1.93 (11) |
| **Hypothermia** | 1.05 (6) |
| **SGA** | 1.05 (6) |
| **Other**[*] | 5.10 (29) |
| **Missing** | 0.70 (4) |

[*] Anaemia, Birth asphyxia, Convulsion, Electrolyte imbalance, Esophageal atresia, Failure to thrive, Grunting, Hypoxic-ischemic encephalopathy, Hydrocephalus, Hyperthermia, Hypoglycaemia, MBA, Non-nutritive sucking, Patent ductus arteriosus failure, Pneumonia, Unable to suck, Weight loss

respiratory distress syndrome (RDS) (4.57%), sepsis (3.34%), extremely low birthweight (ELBW) (2.11%), jaundice (1.93%), hypothermia (1.05%), and small for gestational age (SGA) (1.05%). The remaining 29 neonates (5.10%) had one of the other 15 neonatal conditions, which were categorized as "Other" and detailed under (Table 2). There were 214 infants (37.35%) that were admitted to the KMC room with two of the listed neonatal conditions, 31 (5.41%) with three, and 3 (0.52%) with four concurrent neonatal conditions.

There were 45 neonates (7.85%) omitted from further analysis due to missing data. (Table 3) shows the changes in the weight category for the 528 study participants by comparing their weight at admission and discharge, and (Table 4) shows the median stay and weight gain among participants in the same category. Normal weight for newborns admitted to the KMC room was defined as an infant weighing 2,500 g and above. Neonates weighing below 2,500g upon admission to the KMC room were classified in three subcategories: low weight (LW: 1,500g - 2,499g); very low weight (VLW: 1,000g - 1,499g); and extremely low weight (ELW: <1,000g).

## Effectiveness

The intervention was evaluated using data collected on individual newborn outcomes at admission and upon discharge. We primarily report on weight gain, the primary cause of NICU admission and length of stay related to the KMC model. We assessed the infants' weight gain during their stay in the KMC room. Throughout the study, only two infants (0.35%) died,

**Table 3. Weight categories among neonates upon KMC admission and discharge.**

| Admission Weight Category | Discharge Weight Category | | | |
|---|---|---|---|---|
| | Group A (1,000g – 1,499g) | Group B (1,500g – 2,499g) | Group C (≥2,500g) | Total |
| ELW | 13 | 0 | 0 | **13** |
| VLW | 290 | 129 | 0 | **419** |
| LW | 1 | 77 | 11 | **89** |
| Normal | 0 | 1 | 6 | **7** |
| **Total** | **304** | **207** | **17** | **528** |

**Table 4. Impact of length of stay on weight gain among neonates.**

| Admission Weight Category | Days in KMC (Median, IQR) | Weight Gain (g) (Median, IQR) | Weight Gained per day (g) (Median, IQR) |
|---|---|---|---|
| ELW | 13 (6–19) | 500 (500–550) | 36.67 (26.32–66.67) |
| VLW | 7 (5–12) | 280 (200–300) | 33.33 (18.18–50.00) |
| LW | 6 (4–10) | 100 (100–200) | 20.00 (9.09–40.00) |
| Normal | 9 (3.5–11.5) | 100 (0–200) | 10.00 (0.00–17.50) |

and no nosocomial infections were recorded during infants stay in the KMC room. Thirty infants (5.24%) with conditions that exacerbated and required specialized care during their stay in the KMC room were immediately referred back to the NICU.

Upon admission to the KMC room, 13 neonates (2.46%) were classified as ELW, 419 (78.98%) were classified as VLW, 89 (16.86%) considered LW, and 7 (1.33%) were normal weight ($\geq$2,500g). At discharge, all 13 ELW infants had graduated to Group A weight category (1,000g – 1,499g) showing a median weight gain of 500 g (IQR: 500–550) over a median stay of 13 days (IQR: 6–19). Of the 419 VLW neonates at admission, 290 (69.21%) remained in the same weight category—Group A at discharge while 129 (30.79%) improved to the Group B weight category (1,500g - 2,499g). Infants considered VLW showed a median weight gain of 280 g (IQR: 200–300) over a median stay of 7 days (IQR: 5–12). Among the 89 LW neonates at admission, 1 (1.12%) regressed to the Group A weight category at discharge, 77 (86.52%) remained in the Group B weight category, and 11 (12.36%) improved to Group C weight category ($\geq$2,500g), individually gaining a median of 100 g (IQR: 100–200) over a median stay of 6 days (IQR: 4–10). The 7 (1.33%) neonates who were admitted to the KMC room at normal birthweight stayed a median of 9 days (IQR: 3.5–11.5) and individually gained a median of 100 g (IQR: 0–200), with 6 infants (85.71%) remaining at normal weight and 1 (14.29%) regressing to the Group B weight category. Chi-square and Kruskal-Wallis tests indicated that the differences in median length of stay and weight gain between participants in different birthweight categories among the sample were all significant ($p < 0.05$).

The health outcomes of the neonates included in the study are detailed in (Table 5). The median weight at KMC admission was 1,200 g (IQR: 1100–1390), and the median weight at discharge was 1,400 g (IQR: 1400–1560). The median weight gain for all infants was 200 g (IQR: 150–300). A Kruskal-Wallis test revealed that there were no statistical differences in weight gain based on the infant's exposure to HIV ($p > 0.05$), but there was a statistically significant difference based on the primary cause of NICU admission ($p < 0.05$). Infants admitted to the NICU due to ELBW showed the greatest median weight gain of 400 g (IQR: 400–400), those admitted with VLBW or hypothermia both showed the next greatest median weight gain of 300 g (IQR: 200–400). Neonates admitted to the KMC room who were SGA showed a median weight gain of 250 g (IQR: 200–300), prematurity showed a median weight gain of 200 g (IQR: 160–300), neonates admitted with LBW, RDS, or one of the other 15 conditions exhibited weight gain of 200 g (IQR: 100–300), those admitted with sepsis showed a smaller weight gain of 150 g (IQR: 100–300), and neonates with the primary neonatal condition of jaundice showed the least improvement with a median weight gain of 100 g (100–200).

Dividing the days' neonates spent in the KMC room into quantiles, a Kruskal-Wallis test showed a significant difference ($p < 0.05$) in weight gain depending on the length of stay. Infants in the first quantile (0–4 days) exhibited the greatest median weight gain of 500 g (IQR: 500–550), those in the second quantile (5–7 days) showed the next highest median weight gain of 280 g (IQR: 200–300), the third quantile (8–11 days) showed the smallest median weight gain of 100 g (IQR: 100–200), and those in the fourth quantile (12 or more days) also experienced the smallest median weight gain of 100 g (IQR: 0–200).

**Table 5. Weight gain among neonates admitted to the KMC room (n = 528).**

| | Median (IQR) | P-value |
|---|---|---|
| **Weight at KMC admission (g)** | 1200 (1100–1390) | |
| **Weight at KMC discharge (g)** | 1400 (1400–1560) | |
| **Weight gain at KMC (g) (based on primary cause of NICU admission)** | 200 (150–300) | < 0.05 |
| Prematurity | 200 (160–300) | |
| ELBW | 400 (400–400) | |
| VLBW | 300 (200–400) | |
| LBW | 200 (100–300) | |
| RDS | 200 (100–300) | |
| Sepsis | 150 (100–300) | |
| Jaundice | 100 (100–200) | |
| Hypothermia | 300 (200–400) | |
| SGA | 250 (200–300) | |
| Other | | |
| **Days in KMC** | 200 (100–300) | < 0.05 |
| 0–4 Days | 500 (500–550) | |
| 5–7 Days | 280 (200–300) | |
| 8–11 Days | 100 (100–200) | |
| 12 + Days | 100 (0–200) | |
| **Weight Gained per day at KMC (g)** | 30.00 (16.67–50.00) | > 0.05 |
| **Days in KMC** | | < 0.05 |
| 0–4 Days | 66.67 (33.33–100.00) | |
| 5–7 Days | 40.00 (28.57–57.57) | |
| 8–11 Days | 27.27 (18.18–36.36) | |
| 12 + Days | 15.39 (9.52–22.73) | |
| **Days in KMC Unit (based on primary cause of NICU admission)** | 7 (4–11) | < 0.05 |
| Prematurity | 7 (5–11) | |
| ELBW | 8 (5–11) | |
| VLBW | 9 (4–14) | |
| LBW | 7 (5–11) | |
| RDS | 6 (2–9.5) | |
| Sepsis | 7 (6–10) | |
| Jaundice | 10.5 (7–15) | |
| Hypothermia | 8.5 (7–13) | |
| SGA | 6.5 (4–8) | |
| Other | 4 (2.5–8) | |

## Adoption

The KMC model successfully trained all government nurses in the UTH KMC room in KMC techniques and clinical data management. Following study implementation, the Zambian MoH formally endorsed KMC services and embedded them within the standard of care for preterm and/or LW newborns.

## Implementation

The KMC model focussed on overall KMC implementation at two levels:- tertiary hospital (UTH) and community level in collaboration with the government nurses and SMAGs, respectively. Both levels played a key role in the care of the preterm and/or LW infant. The

study QI nurse's role was to train both the government nurses and mothers in KMC techniques and monitor and collect clinical data during the day in the KMC room at UTH. Before discharge from the NICU, mothers were allowed to practice intermittent KMC during feeding time for 30 minutes to 1 hour daily as clinically unstable infants could only accommodate short periods of KMC. Eligible mother-infant pairs who were recommended for KMC care and further monitoring within the hospital were transferred from the NICU to the KMC room for KMC initiation. Upon admission into the KMC room, the study QI nurse documented mother and the infant's demographic and clinical data. The government nurses worked alongside the study QI nurse for 24 hours a day and received daily mentorship on supporting mothers to initiate and sustain KMC.

Mothers were taught the benefits of KMC, appropriate KMC techniques, and how to securely position their newborns during breastfeeding and skin to skin contact through one on one talks, educational videos, and IEC materials. Mothers secured their newborns clothed in diapers, hats and socks in upright kangaroo positions, skin to skin between their breasts, using convenient wraps from their homes called 'chitenges'. Mothers were encouraged to practice continuous skin to skin contact for at least 20 hours a day throughout their stay, except during diaper changes, feeding, routine clinical assessments, sleeping or using the restroom. The study QI nurse ensured that the KMC room was at optimum temperature using room heaters and supportive to managing hypothermia while mothers practised continuous skin to skin contact throughout the day. Both nurses trained mothers on the importance of practising good hygiene to prevent infections, including handwashing after using the restroom, changing diapers, and expressing breastmilk. Both nurses were stationed in the KMC room and were available to provide individual one-on-one support to the mothers in caring for their newborns and effectively initiating breastfeeding. Cup or breastmilk feeding was recommended every 2 hours daily, with milk ranging from 5 to 15 mls per feed. The frequency and amount of breastmilk a mother expressed depended on the quantity of milk the baby could tolerate per feed and the daily recommended amount based on the gestation age and weight of the newborn. Mothers were taught how to cup feed babies who were unable to breastfeed using the nifty feeding cups by manually expressing the appropriate quantities of breastmilk into the cup and then feeding the newborn. Similarly, the hospital nutritionist counselled mothers that were unable to express breastmilk.

The study QI nurse and government nurses monitored newborns' weight gain, milk intake and observed their vital signs daily. Upon admission to the KMC room, the infant's weight was checked every morning at 6:00 AM throughout their stay and at discharge from the KMC room. The standard discharge weight for newborns in the KMC room was 2500g; however, due to the high patient volume, inadequate space, and increase in the demand for KMC services, newborns who met the facility discharge criteria were discharged from the KMC room earlier. Mothers who expressed confidence in practising KMC with their clinically stable newborns who fed well, gained 15g or more for three consecutive days, and weighed at least 1500g were discharged home from the KMC room. However, if a baby was awaiting admission to the KMC room, exceptions were made to discharge babies below 1,500grams but met the other discharge criteria. Conversely, newborns who gained less than 10g per day for three consecutive days, lost weight, or exhibited danger signs in their vitals were immediately referred back to the NICU for specialized care and treatment.

Additionally, we assessed mother-infant pairs' readiness to be discharged home through a KMC daily scoring chart (S1 Appendix) for three consecutive days. It consisted of 10 questions that assessed how the newborn had been in the past three days and each response was ranked on a scale from 0 to 2. The total score varied from a minimum of 0 to a maximum of 60. Breastfeeding mother-infant pair scores of 19 and above were considered ready to be discharged,

while low scores on the chart indicated the need for additional attention and care. We assessed the following according to the three KMC components:- 1) acceptance and application of KMC and knowledge of KMC 2) baby positioning and attachment to the breast; mother's milk production; baby's ability to suckle at the breast; confidence in handling baby and baby's weight gain per day 3) confidence in caring baby at home and social economic support.

Upon discharge, mothers were encouraged to practice continuous KMC at home until the baby weighed 2500g or was uncomfortable being in the KMC position. The government community Safe Motherhood Action Groups (SMAGs) followed up mothers at their homes within 2–3 days after discharge as part of routine patient care to ensure that they adjusted to the home environment, applied newborn care practices, and were breastfeeding adequately. Additionally, mothers were asked to share their challenges of home care post-discharge and were counselled on how to adequately overcome those challenges. The subsequent follow-up visits were then scheduled on a weekly basis until the baby gained 3 kgs and was discharged for bacilli Calmette-Guerin vaccine and other immunization vaccinations. In most circumstances, follow up visits were dependent on the mother and baby's needs and level of social support at the household. If the baby exhibited danger signs at a follow-up visit, the baby was immediately referred to the NICU at UTH.

### Maintenance

Following the study closeout, the government nurses took over from the study QI nurse and continued providing KMC services at UTH. Due to the demand created by our study, with support from the hospital administration, the KMC room was eventually relocated to a larger room within the hospital, increased its bed spaces from 10 to 26, implemented the predischarge readiness scoring system, and hired six additional nurses, further decongesting the NICU. SMAGs who were not fully utilised at the community level have been identified as key MoH support staff to be trained in KMC, strengthening the continuum of care from the hospital to the community. Furthermore, in 2018 MoH, in close partnership with stakeholders, adapted the 2015 WHO KMC guidelines launching the Kangaroo Mother Care National Guidelines and distributed them nationwide in 2019. This guideline has paved the way for the roll-out and uptake of KMC services across Zambia. KMC rooms have since been established in four high-volume general hospitals in four provincial districts in Zambia, namely in Livingstone, Mansa, Kabwe, and Chipata districts, with more underway. Additionally, the PREEMI and the Saving Mothers, Giving Life programs jointly adopted and implemented the KMC model in seven district hospitals in Zambia.

### Discussion

Similar to other studies focused on hospital-based implementation of KMC, this prospective descriptive cohort study showed that KMC positively impacted the weight gain of newborns admitted to the KMC room [34, 35]. Each RE-AIM dimension provided valuable information on the impact of the KMC model at an individual and organisational level. At an individual level, we demonstrated that all infants enrolled in the study participated in KMC, and almost all showed varying degrees of increased weight. Through engaging multiple audiences, including mothers, health care workers and SMAGs our model's reach, grew significantly larger and expanded within the hospital to additional district and provincial level hospitals. We noted that the successful implementation and scale up of the KMC model in a tertiary level hospital is likely to be achieved through the provision of supplies, equipment, routine data quality management and daily onsite coaching and mentorship of MoH staff. These findings were echoed by a study from Ethiopia and India that suggested that scaling up context adapted KMC

models such as our KMC model, when supported by health care workers and the government in health facilities, show high coverage across large populations [18].

Despite the several barriers identified in facility-based implementation of KMC in literature, such as lack of adequate staff, resources and community structures, our model was introduced into routine standard of care and leveraged student nurses at UTH [36, 37]. Our findings suggest that student nurses serving as champions, in settings where experienced staff are unavailable can ensure longevity of KMC at an organisational level. Several studies in multiple countries support the use of student nurses in providing quality patient care in hospitals [38, 39]. While UTH has a KMC unit with a dedicated seasoned nurse, other higher-level facilities in similar settings can replicate the use of student nurses to ensure the uptake and scale-up of KMC services. Furthermore, to improve the overall efficacy and sustainability of KMC, systematic support should be present. KMC should be institutionalized and incorporated into the healthcare system with a champion encouraging and promoting the practice at both health facility and community level. Democratic Republic of Congo (DRC), Uganda, Malawi, and Kenya are a few countries in the region that have implemented KMC nationally and incorporated it into their standard of care [40].

Thirdly, healthcare systems will need to work to identify and remove any barriers that may prevent the implementation of KMC. Gender role stereotypes have been noted as obstacles to this practice [14], with childcare often viewed as a mother's responsibility after they return home [41]. Although there is not much research on male involvement in KMC, findings from studies conducted in Ethiopia and Ghana suggest that patriarchal societies often affect mothers sustenance of KMC in the hospital [42–44]. Our study, however, noted several men practising KMC at home following sensitization to family members before hospital discharge. To address the stigma associated with male involvement in KMC men known to practice KMC at community level could be used to serve as mentors, as a potential strategy for KMC scale up. Additionally, the name Kangaroo Mother Care may give the impression that the intervention is only a mother's duty and could be revised to promote the practice in a more inclusive light [45].

Furthermore, KMC will need continued support from all healthcare system levels, including both government and non-government stakeholders [14]. Nurses need to act as advocates for KMC, and the administration may need to employ SMAGs to ensure the continuity of KMC in the community [16]. SMAGs can assist in identifying infants delivered outside of a health facility that may benefit from KMC. In rural Tanzania and Uganda, SMAGs identified these infants and facilitated the necessary referrals [40]. Follow up mechanisms need to be strengthened for infants discharged from health facilities due to limited bed space requiring KMC [5]. Early and consistent follow up of these infants can help monitor KMC at the community level and provide vital information on the infant's overall health. The constant follow up may also motivate mothers to continue the process of KMC at household level.

This study has several key strengths. First, the study collected clinical data on all infants from admission to the KMC room and their discharge. Weight was measured upon admission to the KMC room and upon discharge, with weight gains monitored for 99% of the enrolled infants. Second, the study also collected data on feeding practices, which showed that all infants were either breastfed or given breastmilk using the Nifty feeding cups (Laerdal Global Health, Stavanger, Norway). Third, the study will provide learning findings on the importance of strengthening data management in low resource-limited settings at the health facility level. Our study includes a complete dataset on all study participants from admission to discharge, with data on NICU readmissions and no cases of loss to follow up. These findings can serve as baseline data for future KMC evaluation studies in Zambia that focus on weight gain as a neonatal outcome. Lastly, these infants were followed up after their discharge from KMC for monitoring for any adverse events that would warrant readmission to NICU, KMC, or other units

at the hospital. Follow up enabled us to enumerate readmissions in this cohort. Given that we implemented our KMC model in the largest and main tertiary level hospital, the findings of this study may help the MoH shape policies regarding the scale-up of KMC protocols for newborn care in Zambia.

Our study has several limitations, too. Operationalizing the RE-AIM framework to describe the implementation of the KMC model and their neonatal outcomes proved to be challenging as there were significant gaps in the UTH records for NICU and KMC data before our study. For this reason, we were unable to evaluate its impact on the length of stay in the period before implementing our model. Additionally, while mothers practiced continuous KMC for atleast 20 hours a day throughout their stay, we did not record the actual duration of time mothers spent practising skin to skin. This would have been beneficial in supporting the idea that increased time spent in KMC increases breastfeeding and is directly proportionate to infant's weight gain [35, 46–48]. Secondly, maternal risk factors, such as maternal HIV status, hemoglobin level, or weight, were equally not fully documented in the government registers and therefore could not be used to surmise the infant's ability to gain weight. Similarly, data on birthweight and the date of birth of the infants was not collected. Weight information at admission and following discharge from the KMC room was missing for 45 infants, which could have impacted the statistical analysis. Thirdly, we did not collect qualitative data on the barriers and facilitators to the adoption of KMC study from the health care workers and mothers in the hospital and community. This information would be critical in assessing the impact of our KMC model, their motivation to overcome any barriers and understanding the different views of satisfaction with our model. Similarly, details on the overall costs of the KMC model were also not assessed. Understanding the cost of implementing the model in tertiary level hospitals is essential to key decision makers prioritising and sustaining KMC in communities [49]. Lastly, we did not collect household data on the duration of KMC practice and weight gain as follow up, monitoring, and referral from the community was organised and led by the SMAGs. Furthermore, ideally, infants should have been followed up for longer periods post-discharge (i.e. 1 year of age) to determine long-term effects of KMC, such as weight gain or occurrence of disease or illness.

## Conclusions

While implementing KMC in the tertiary level hospital in Lusaka, Zambia was a challenging and ambitious undertaking, after one year there was sufficient evidence of its uptake by the healthcare providers and mothers and of its effect on weight gain among preterm and low birthweight infants. National KMC coverage in Zambia remains low, thus we recommend that the KMC implementation model is replicated for routine use in health facilities for all stable babies <2000 g. Additionally, KMC should be incorporated into all relevant programs including Essential Newborn care and integrated as part of the continuum of maternal, newborn, and child care, which could substantially reduce neonatal mortality among preterm births in Zambia.

## Supporting information

**S1 Appendix. KMC predischarge readiness scoring chart.**
(DOCX)

## Acknowledgments

We would like to thank the Ministry of Health, particularly the government staff working at the UTH KMC and NICU who assisted with the overall study implementation and patient monitoring. Furthermore, we would like to thank the members of the PREEMI study team,

particularly the study QI nurse who worked with the mothers training them in KMC and monitored their babies throughout the duration of the study.

## Author Contributions

**Conceptualization:** Colm Travers, J. Anitha Menon, Albert Manasyan.

**Data curation:** Branishka Lewis, Keilya Kasprzyk, Aaron Mangangu, Herbert Kapesa, Albert Manasyan.

**Formal analysis:** Branishka Lewis, Keilya Kasprzyk, Herbert Kapesa.

**Funding acquisition:** Albert Manasyan.

**Investigation:** Albert Manasyan.

**Methodology:** Martha Mwendafilumba, Albert Manasyan.

**Project administration:** Nobutu Muttau, Martha Mwendafilumba, Kunda Mutesu-Kapembwa, Albert Manasyan.

**Supervision:** Kunda Mutesu-Kapembwa, Albert Manasyan.

**Validation:** Kunda Mutesu-Kapembwa, Albert Manasyan.

**Visualization:** Herbert Kapesa.

**Writing – original draft:** Nobutu Muttau, Albert Manasyan.

**Writing – review & editing:** Nobutu Muttau, Martha Mwendafilumba, Branishka Lewis, Keilya Kasprzyk, Colm Travers, J. Anitha Menon, Kunda Mutesu-Kapembwa, Aaron Mangangu, Herbert Kapesa, Albert Manasyan.

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
