## [Decision Letter · Decision Letter 0]

28 Apr 2022

PONE-D-22-01178Implementation of a Kangaroo Mother Care Model at a Tertiary Level Hospital in Zambia Using the RE-AIM FrameworkPLOS ONE

Dear Dr. Manasyan,

Thank you for submitting your manuscript to PLOS ONE. After careful consideration, we feel that it has merit but does not fully meet PLOS ONE’s publication criteria as it currently stands. Therefore, we invite you to submit a revised version of the manuscript that addresses the points raised during the review process.

We look forward to receiving your revised manuscript.

Kind regards,

Hannah Tappis, DrPH, MPH

Academic Editor

PLOS ONE

Journal Requirements:

2. Please include a complete copy of PLOS’ questionnaire on inclusivity in global research in your revised manuscript. Our policy for research in this area aims to improve transparency in the reporting of research performed outside of researchers’ own country or community. The policy applies to researchers who have travelled to a different country to conduct research, research with Indigenous populations or their lands, and research on cultural artefacts. The questionnaire can also be requested at the journal’s discretion for any other submissions, even if these conditions are not met.  Please find more information on the policy and a link to download a blank copy of the questionnaire here: https://journals.plos.org/plosone/s/best-practices-in-research-reporting. Please upload a completed version of your questionnaire as Supporting Information when you resubmit your manuscript

3.  This RA mentioned a Clinical Trial registration. Please check whether CT checks should follow.

5. We note that you have stated that you will provide repository information for your data at acceptance. Should your manuscript be accepted for publication, we will hold it until you provide the relevant accession numbers or DOIs necessary to access your data. If you wish to make changes to your Data Availability statement, please describe these changes in your cover letter and we will update your Data Availability statement to reflect the information you provide

Reviewers' comments:

Reviewer's Responses to Questions

**Comments to the Author**

1. Is the manuscript technically sound, and do the data support the conclusions?

Reviewer #1: Yes

Reviewer #2: Yes

2. Has the statistical analysis been performed appropriately and rigorously? 

Reviewer #1: Yes

Reviewer #2: No

3. Have the authors made all data underlying the findings in their manuscript fully available?

Reviewer #1: No

Reviewer #2: No

4. Is the manuscript presented in an intelligible fashion and written in standard English?

Reviewer #1: Yes

Reviewer #2: Yes

5. Review Comments to the Author

Reviewer #1: Thank you for the opportunity to review the diligent work of Muttau et al, Implementation of a Kangaroo Mother Care Model at a Tertiary Level Hospital in Zambia Using the RE_AIM Framework.”

I recommend consideration for publication after minor revision.

The authors are to be congratulated on collecting such important data on KMC intervention in an environment with a high burden of neonatal mortality. The authors report an impressive weight gain and an acceptable average hospital stay. The information in this study will guide clinicians in resource limited settings, to use such models for quality improvement in Kangaroo Mother Care methods. I have several questions and suggestions intended to clarify and strengthen the article.

Comments:

1. I believe that the methods of data collection was appropriate although the study is missing a comparison period which to me is a significant missing opportunity to evaluate the model appropriately. A period of initial baseline data collection even for 3 months would have been enough to make a conclusion.

2. The time spent skin-to-skin is important for maximizing the benefits of KMC. The fact that this was not documented is also another limitation (mentioned by the authors as well) , although I acknowledge the difficulties of obtaining this information.

3. Overall, what is the newborn mortality rate in this hospital with 20,000 deliveries per year. Furthermore, it will be important to understand preterm birth rates in this hospital to understand the magnitude or the needs for KMC facilities. Where were the other preterm being taken care of if there was no space in Kangaroo ward (before expansion)

4. The author should explain what happened (outcome) to the 28 newborns who went back to NICU (Line 189-191). The conditions that necessitated them to go back should also be clearly stated.

5. The authors concluded that there was evidence that the model improved neonatal outcomes. It is difficult to justify this statement and I recommend to change it (Line 457)

6. Line 293 I agree this is a quality improvement initiative, however, as researchers, it is good practice to ask for consent for using data for publication. Verbal consent would have sufficed.

Reviewer #2: This is a very relevant and priority area of research in the context of LMICs with high proportions of LBW and preterm babies, inadequate access to appropriate facility care, sub-optimally functioning government health facilities and the challenge of resource allocation with multiple competing interest. The study addresses an important question of KMC implementation in facility.

Few issues that may be reviewed to strengthen the manuscript

This study has been based on the principles of implementation research (IR) as it appears from the manuscript. RE-AIM framework has been used but complete description is lacking. The first dimension of RE-AIM, i.e., Reach has not been addressed. It is the penetration of the project into its intended audience, in this case it could have been the LBW and preterm newborns in the defined study area. IR aims at population-based coverage. This study was conducted in one facility, in that context Reach could be defined as the proportion of all LBW and/or preterm babies who received KMC out of all such babies born or referred to during the study period.

Ideally, the intervention should have been institutionalized to address the Maintenance component of RE-AIM, both at the facility and community levels.

For scaling up any efficacious or effective intervention, cost is an important information. The authors can state if they are planning to or have already conducted any economic analysis.

A relevant reference can be added, Mony PK et al., Scaling up Kangaroo Mother Care in Ethiopia and India: a multi-site implementation research study. BMJ Glob Health. 2021 Sep;6(9): e005905.

Specific comments

Line 125: stable babies are hemodynamically stable, need to mention whether they were on IV fluids or not or whether they were able to feed orally.

Lines 135: only one study nurse was engaged 24*7 and 7 days a week, this does not seem feasible, the authors can elaborate on this. What was the back up for leaves?

Were any pre-facility interventions implemented to refer all eligible infants to the hospital

The eligibility criteria for KMC initiation need to be specified- age of the infant, birth weight, gestational age, etc. The mean (SD) age of initiation needs to be mentioned. The mean (SD) gestational age in weeks of the enrolled infants need to be mentioned. The duration of skin to skin contact over 24 hours and in days is missing. What was the definition of KMC in this IR, did the definition include hours of skin to skin contact over 24 hours and exclusive breast feeding? If yes, data needs to be shown in the manuscript. Analysis to determine the association of weight gain with the hours of skin to skin contact (dose-response) would be an important information. The exact period of follow up is not mentioned. The authors did not mention anything about sample size calculation.

There is no description of the study teams. IR always involves mixed methods design with a qualitative component. There is no mention of this. It is not clear whether the outcome measurement team was independent form the persons supporting implementation.

It is not clear why normal birth weight infants were admitted to KMC unit, KMC is an intervention for LBW and/or preterm babies, if these babies were preterm, that needs to be mentioned.

Table 3a shows the total number of babies as 528 but the sample enrolled was 573.

The authors have described well the strengths and limitations in the discussion section. The discussion should also include a section on research in context and policy implications.

6. PLOS authors have the option to publish the peer review history of their article (what does this mean?). If published, this will include your full peer review and any attached files.

Reviewer #1: No

Reviewer #2: **Yes: **Sarmila Mazumder

---

## [Author Response · Author response to Decision Letter 0]

11 Jul 2022

Emily Chenette, 

Editor-in-Chief

PLOS One Journal

USA

10 June 2022

Dear Ms. Chenette,

RE: SUBMISSION OF REVISED MANUSCRIPT - PONE-D-22-01178

On behalf of my co-authors, I would like to thank you for providing us with an opportunity to respond to the comments raised by the reviewers for the manuscript retitled, “Strengthening Kangaroo Mother Care at a Tertiary Level Hospital in Zambia: A prospective descriptive study" (former - Implementation of a Kangaroo Mother Care Model at a Tertiary Level Hospital in Zambia: a descriptive study using the RE-AIM framework). 

We have reviewed each comment carefully and have made the necessary changes to further our manuscript towards publication. We have enclosed two versions of the revised manuscript in Word format (.docx)—one with changes highlighted under “track changes” and a second “clean” version with all changes accepted. While reviewing the manuscript, we have made additional formatting (cover page, IRB, and Abbreviations sections) and context (addition of new findings and references, grammer, flow) to the manuscript, which too have been tracked in the manuscript.

Research reported in this publication was supported by a grant from The ELMA Foundation (14-F0023). The funders had no role in study design, data collection and analysis, decision to publish, or preparation of the manuscript. Authors N. Muttau, M. Mwendafilumba, A. Mangangu, H. Kapesa, and A. Manasyan received salary support from this donor during the implementation of this study. 

All data generated from this study are owned by the Ministry of Health of the Republic of Zambia thus will be shared upon request from the corresponding author or Dr Sylvia Machona-Muyunda (Head of the NICU, University Teaching Hospital, Ministry of Health: sylviammm@yahoo.com).

Please find below our point-by-point responses to all the comments accompanied by the revised manuscript with tracked changes. 

We look forward to your consideration.

Sincerely,

Albert Manasyan, MD, MPH

Head of Department

Reproductive, Maternal, Newborn, and Child Health 

Centre for Infectious Disease Research in Zambia (CIDRZ)

Reviewer #1: Thank you for the opportunity to review the diligent work of Muttau et al, Implementation of a Kangaroo Mother Care Model at a Tertiary Level Hospital in Zambia Using the RE_AIM Framework.”

I recommend consideration for publication after minor revision.

The authors are to be congratulated on collecting such important data on KMC intervention in an environment with a high burden of neonatal mortality. The authors report an impressive weight gain and an acceptable average hospital stay. The information in this study will guide clinicians in resource limited settings, to use such models for quality improvement in Kangaroo Mother Care methods. I have several questions and suggestions intended to clarify and strengthen the article.

Comments:

1. I believe that the methods of data collection was appropriate although the study is missing a comparison period which to me is a significant missing opportunity to evaluate the model appropriately. A period of initial baseline data collection even for 3 months would have been enough to make a conclusion.

Response: Thank you for your suggestion. We planned to do pre-post analysis, however, due to poor health record management of hospital-level NICU and KMC data for the previous year, we are only able to present these findings within the realm of our collected data. Our findings will serve as baseline data for researchers and program implementers looking to strengthen KMC programs at both facility and community level in similar resource limited settings. We realize that a comparison period would have made our manuscript stronger and have acknowledged it as a study limitation (lines 692 - 696).

2. The time spent skin-to-skin is important for maximizing the benefits of KMC. The fact that this was not documented is also another limitation (mentioned by the authors as well) , although I acknowledge the difficulties of obtaining this information.

Response: Thank you for your comment. We have revised out statement to indicate that mothers practices continuous KMC throughout their stay in the KMC room, except when attending to personal needs (lines 503 - 506). 

3. Overall, what is the newborn mortality rate in this hospital with 20,000 deliveries per year. Furthermore, it will be important to understand preterm birth rates in this hospital to understand the magnitude or the needs for KMC facilities. Where were the other preterm being taken care of if there was no space in Kangaroo ward (before expansion).

Response: Thank you for your suggestion. We have revised the statement to include the preterm birth and institutional mortality rates during that period (lines 174 - 175). We have also included the number of serviceable incubators at the time (line 176).

4. The author should explain what happened (outcome) to the 28 newborns who went back to NICU (Line 189-191). The conditions that necessitated them to go back should also be clearly stated.

Response: Thank you for your comment. Unfortunately, we did not collect outcome data on newborns who were readmitted to the NICU. Although we do not have specific reasons for their readmission to the NICU based on the general information, infants were commonly known to suffer from respiratory distress syndrome, sepsis, jaundice and hypothermia upon initial admission to the NICU (lines 366 - 371).

5. The authors concluded that there was evidence that the model improved neonatal outcomes. It is difficult to justify this statement and I recommend to change it (Line 457).

Response: Thank you for your suggestion. We have since revised the statement to indicate its impact on weight gain versus overall neonatal outcomes (line 729 - 732). 

6. Line 293 I agree this is a quality improvement initiative, however, as researchers, it is good practice to ask for consent for using data for publication. Verbal consent would have sufficed.

 Response: Thank you for your comment. We were given a waiver to collect informed consent by both the local (UNZA-BREC and NHRA) and international (UAB-IRB) regulatory authorities given that KMC was standard of care. Furthermore, prior to publication, we were given approval by the Zambian Ministry of Health to publish these study findings (lines 317 - 322). However, as part of medical practice the nurses upheld their ethical and legal responsibility of informing the mothers about KMC before admitting them into the KMC room. 

Reviewer #2: This is a very relevant and priority area of research in the context of LMICs with high proportions of LBW and preterm babies, inadequate access to appropriate facility care, sub-optimally functioning government health facilities and the challenge of resource allocation with multiple competing interest. The study addresses an important question of KMC implementation in facility.

1. Few issues that may be reviewed to strengthen the manuscript

This study has been based on the principles of implementation research (IR) as it appears from the manuscript. RE-AIM framework has been used but complete description is lacking. The first dimension of RE-AIM, i.e., Reach has not been addressed. It is the penetration of the project into its intended audience, in this case it could have been the LBW and preterm newborns in the defined study area. IR aims at population-based coverage. This study was conducted in one facility, in that context Reach could be defined as the proportion of all LBW and/or preterm babies who received KMC out of all such babies born or referred to during the study period.

Response: Thank you for your comment. We have revised the manuscript to include a description and a figure of the RE-AIM framework in the methods section (lines 281 - 294). Additionally, we have now indicated that the proportion is the number of newborns discharged from the NICU and eligible to be admitted to the UTH KMC room (lines 328 - 339). We have also provided justification for why UTH served as the ideal study site to provide population-based coverage (lines 169 - 173).

2. Ideally, the intervention should have been institutionalized to address the Maintenance component of RE-AIM, both at the facility and community levels.

Response: Thank you for your comment. KMC was adopted by the Zambian Ministry of Health, which ensured its maintenance at both health facility and community levels through close cooperation with government nurses and SMAGs respectively (lines 590 - 597). 

3. For scaling up any efficacious or effective intervention, cost is an important information. The authors can state if they are planning to or have already conducted any economic analysis.

Response: Thank you for your suggestion. We were not able to conduct a cost-effectiveness analysis due to lack of available resources at the time of the study (lines 719 - 722).

4. A relevant reference can be added, Mony PK et al., Scaling up Kangaroo Mother Care in Ethiopia and India: a multi-site implementation research study. BMJ Glob Health. 2021 Sep;6(9): e005905.

Response: Thank you for your suggestion. We have added the reference to our manuscript (reference #18).

Specific comments

5. Line 125: stable babies are hemodynamically stable, need to mention whether they were on IV fluids or not or whether they were able to feed orally.

Response: Thank you for your comment. We have revised the statement to indicate that babies who did not require respiratory support, were hemodynamically stable, and feeding orally were admitted into the KMC room (line 209).

6. Lines 135: only one study nurse was engaged 24*7 and 7 days a week, this does not seem feasible, the authors can elaborate on this. What was the back up for leaves?

Response: Thank you for your comment. We have revised the statement to indicate that the study hired QI Nurse worked with several government nurses in ensuring 24/7 coverage in the KMC room (lines 233 - 236). 

7. Were any pre-facility interventions implemented to refer all eligible infants to the hospital

The eligibility criteria for KMC initiation need to be specified- age of the infant, birth weight, gestational age, etc. 

Response: Thank you for your comment. As KMC was already part of the standard of care at UTH, we did not introduce new interventions to the facility, but only trained the mothers admitted to the KMC room. Similarly, following hospital protocol, we utilized their inclusion criteria for admission to the KMC room (lines 207 - 211). 

8. The eligibility criteria for KMC initiation need to be specified- age of the infant, birth weight, gestational age, etc.The mean (SD) gestational age in weeks of the enrolled infants need to be mentioned. 

 Response: Thank you for your comment. Unfortunately, due to incomplete health record management at the NICU, we were not able to present gestational age data in our manuscript (lines 696 -698).

9. The duration of skin to skin contact over 24 hours and in days is missing. 

 Response: Thank you for your comment. Unfortunately, we did not collect data on the actual duration of time mothers spent practicing skin to skin. However, mothers practiced continuous KMC for at least 20 hours daily throughout their stay except during diaper changes, feeding, routine clinical assessments, sleeping or when using the restroom (lines 503 - 506).

10. What was the definition of KMC in this IR, did the definition include hours of skin to skin contact over 24 hours and exclusive breast feeding? If yes, data needs to be shown in the manuscript. 

Response: Thank you for your comment. We have included sentences in the manuscript to explicitly describe KMC, the hours of skin to skin contact and breastfeeding (lines 499 - 523).

11. Analysis to determine the association of weight gain with the hours of skin to skin contact (dose-response) would be an important information. 

Response: Thank you very much for your comment. While the mothers practiced continuous skin to skin, cup or breastmilk feeding was recommended as often as every 2 hours daily with milk amounts ranging from 5 to 15 mls per feed throughout the day (lines 516 - 517). We did not document the exact duration of skin-to-skin. Therefore, unfortunately we are not able to present the impact of duration of skin-to-skin on weight gain.

12. The exact period of follow up is not mentioned. 

Response: Thank you for your comment. The SMAGs followed up mothers at their homes within 2-3 days after discharge as part of routine patient care. The subsequent follow-up visits were then scheduled on a weekly basis until the baby gained 3 kgs and was discharged for bacilli Calmette-Guerin vaccine and other immunization vaccinations (lines 576 - 579).

13. The authors did not mention anything about sample size calculation.

Response: Thank you for your comment. We conducted a prospective descriptive study, embedded in the larger PREEMI trial where we aimed to monitor the KMC practices and its uptake on all admitted mother-infant pairs at UTH between January 2016 and September 2017. We did not set a sample size for this activity (lines 205-207).

14. There is no description of the study teams.

Response: Thank you for your comment. We now explicitly describe the study team and their roles in the methods section (lines 230 – 241) and (lines 298 - 302).

15. IR always involves mixed methods design with a qualitative component. There is no mention of this. It is not clear whether the outcome measurement team was independent from the persons supporting implementation.

Response: Thank you for your comment. The outcome measurements were done by the government nurses and the study QI nurse while the study QI nurse implemented the KMC model (lines 233 - 241). We did not collect qualitative data from healthcare providers and mothers practicing KMC, which we have indicated as a study limitation (lines 715 – 717).

15. It is not clear why normal birth weight infants were admitted to KMC unit, KMC is an intervention for LBW and/or preterm babies, if these babies were preterm, that needs to be mentioned.

Response: Thank you for your comment. Due to overcrowding in the NICU, the KMC room served as an intermediary room between the NICU and home for all infants (including those of normal birth weight) not requiring NICU services but in need of further clinical monitoring within the hospital (lines 177 - 203).

16. Table 3a shows the total number of babies as 528 but the sample enrolled was 573.

Response: Thank you for your comment. We have indicated in the manuscript that 45 neonates were omitted from the analysis due to missing data (line 384).

17. The authors have described well the strengths and limitations in the discussion section. The discussion should also include a section on research in context and policy implications.

Response: Thank you for your comment. We now explicitly describe research in context and policy implications in the discussion section (lines 607 – 676) and (lines 693 - 695).

---

## [Editor Report · Decision Letter 1]

20 Jul 2022

Strengthening Kangaroo Mother Care at a Tertiary Level Hospital in Zambia: A prospective descriptive study

PONE-D-22-01178R1

Dear Dr. Manasyan,

We’re pleased to inform you that your manuscript has been judged scientifically suitable for publication and will be formally accepted for publication once it meets all outstanding technical requirements.

Kind regards,

Hannah Tappis, DrPH, MPH

Academic Editor

PLOS ONE
---

## [Editor Report · Acceptance letter]

22 Aug 2022

PONE-D-22-01178R1 

Strengthening Kangaroo Mother Care at a Tertiary Level Hospital in Zambia: A prospective descriptive study   

Dear Dr. Manasyan:

I'm pleased to inform you that your manuscript has been deemed suitable for publication in PLOS ONE. Congratulations! Your manuscript is now with our production department. 

Kind regards, 

on behalf of

Dr. Hannah Tappis 

Academic Editor

PLOS ONE